# Quantitative Assessment of the Relationship between Land Use/Land Cover Changes and Wildfires in Southern Europe

Joana Parente [1], Marj Tonini [2], Zoi Stamou [3], Nikos Koutsias [3] and Mário Pereira [4,5,*]

1    cE3c—Center for Ecology, Evolution and Environmental Changes & CHANGE–Global Change and Sustainability Institute, Sciences Faculty, University of Lisbon, 1649-004 Lisboa, Portugal; jrparente@fc.ul.pt
2    Institute of Earth Surface Dynamics, Faculty of Geosciences and Environment, University of Lausanne, 1015 Lausanne, Switzerland; marj.tonini@unil.ch
3    Department of Environmental Engineering, University of Patras, G. Seferi 2, GR-30100 Agrinio, Greece; stamou@upatras.gr (Z.S.); nkoutsia@upatras.gr (N.K.)
4    Centre for Research and Technology of Agro-Environmental and Biological Sciences (CITAB), Inov4Agro, University of Trás-os-Montes and Alto Douro (UTAD), Quinta de Prados, 5000-801 Vila Real, Portugal
5    Dom Luiz Institute (IDL), University of Lisbon, 1649-004 Lisboa, Portugal
*    Correspondence: gpereira@utad.pt; Tel.: +351-259-350-728

**Abstract:** Wildfires are key drivers of land use/land cover (LULC) dynamics by burning vegetation and affecting human infrastructure. On the contrary, LULC changes (LULCCs) may affect the fire regime by influencing vegetation type, burnable areas, fuel loads and continuity. This study investigates the relationship between LULCC and wildfires. We developed a methodology based on different indicators, which allowed us to quantitatively assess and better understand the transitions between LULC classes and burnt area (BA) in Europe in the last two decades (2000–2019). The assessment was performed for the entire European continent and, independently, for each of the five European countries most affected by wildfires: Portugal, Spain, France, Italy and Greece. The main results are the following: (i) LULCC analysis revealed a net loss in forests and arable land and a net gain in shrubs; (ii) most of the BA occurred in forests (42% for the whole of Europe), especially in coniferous forests; (iii) transitions from BA generally were to transitional woodland/shrub or, again, to BA. Overall, our results confirm the existence of a strong relationship between wildfires and LULCCs in Europe, which was quantified in the present study. These findings are of paramount importance in fire and environmental system management and ecology.

**Keywords:** wildfires; land use/land cover; burnt area; LULC changes; LULC transitions; Southern Europe

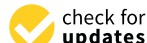



## 1. Introduction

Wildfire is an unplanned and unwanted combustion of dead or alive vegetal biomass in a rural or natural area, often caused by natural causes (e.g., lightning) but mostly by humans, either through negligence or intentionally, that affectd, partially or completely, forest, scrublands and/or agricultural areas [1–4]. Wildfires can be disastrous events causing devastation to the natural environment, infrastructure and affecting wildlife and human health [5].

Wildfires have different environmental and socioeconomic drivers, including climate, weather, topography, vegetation, human factors, fire management strategies and changes in the land use pattern [3,6–9]. Fire regime is defined as the pattern of fire occurrence, frequency, size and severity, as well as land use/land cover (LULC), especially vegetation type and characteristics and fire effects in a given area [4]. Fire regime is changing in almost all the planet's ecosystems [10], mainly as a consequence of climate changes [11–13] and land use/land cover changes (LULCCs) [14,15]. All these changes are expected to increase in the short term due to the highly complex and non-linear interaction between climate,

vegetation, fire regime, human LULC and population [16,17]. While the effect of climate change on wildfires has been largely investigated for many years now [18–21], research at the regional and global scales seeking to characterize and quantify the relationship between burnt area (BA) and LULCCs is lacking in the literature.

Climate change can play a role by leading to changes in land use and management practices [22], with extreme weather events, such as droughts and heat waves, impacting vegetation and increasing fire risk [22–28]. LULCCs in Europe were found to increase the fire occurrence probability in the vicinity of urban areas and human infrastructure because of the anthropogenic origin of most wildfires [29,30]. Despite these findings, there is still a lack of knowledge about the relationship and reciprocal influence between LULCCs and wildfires. Indeed, this investigation could greatly support fire risk assessment, landscape management, prevention activities, detection and suppression systems and recovery strategies. Moreover, although the relationship between LULC and wildfires is generally recognized qualitatively, its quantitative assessment at the European scale is a gap in the state of current knowledge.

To fill this gap, the present study aims to better understand and quantify the transitions between LULC and the BA on the entire European continent and, specifically, in the five most affected Mediterranean European countries in two decades (2000–2019). In more detail, our study seeks to answer the following research questions (RQs):

RQ1: What were the classes of LULC that prevailed and changed most in Europe in the last two decades (2000–2019)? In the context of this RQ, it is hypothesized that: (i) most (more than 50%) of the area of the regions in Europe that underwent LULC changes occurred in forest and scrubland; (ii) this proportion has not changed significantly over the last two decades; (iii) in this period, this proportion did not vary significantly between the countries of Southern Europe; and (iv) the final area of the forest should have decreased and scrubland increased. Additionally, the main objective is to characterize the LULCCs in Europe and, in particular, in the southern countries, while the second is to verify if these ULCCs can be compatible with the occurrence of fires. This explorative analysis is extremely important to help us understand the interactions between humans and natural phenomena, especially in regions with intense social and environmental pressure and where environmental management systems need to be improved (Elena Ana et al. 2013).

RQ2: What were the major transitions in LULCCs that occurred from and to the mapped BAs? Within the scope of this RQ, the research hypotheses were: (i) most of the area of the LULCC transitions that occurred from and to burnt areas occurs mainly in forests and bushes; (ii) most of the area of the LULCC transitions occurring in burnt areas originates from forests, especially conifers; (iii) most of the area of the LULCC transitions occurring in burnt areas gives rise to transitional woodland/shrub or, again, to burnt areas; and (iv) these patterns may vary in southern countries due to their different characteristics (e.g., socioeconomic), despite their similarities (e.g., geographic and climatic). The main objectives of this RQ are to characterize the LULCC transitions in burnt areas and, consequently, to quantify the relationship between LULCCs and fires in Europe and, in particular, in southern countries. The answer to this question might confirm that wildfire is an environmental disturbance factor in many LULC processes, such as vegetation cover (Lavorel et al. 2007), land degradation and soil erosion (Vieira et al. 2015; Parente et al. 2022).

The quantitative assessment of the LULCC within the BA is of fundamental importance to better understand the environmental system dynamics and their interaction with environmental and social drivers. To achieve this objective, only official datasets developed at the European and national scales were selected and used (Table 1), which, due to their high quality, are often used by researchers.

**Table 1.** Datasets used, data providers and spatial resolution.

| Spatial Dataset | Source | Resolution |
|---|---|---|
| CORINE Land Cover Changes (CLCC) | European Environment Agency/Copernicus (https://land.copernicus.eu/pan-european/corine-land-cover, accessed on 1 September 2022). | $250 \times 103\ m^2$ |
| European Forest Fire Information System Burnt Area Dataset (EFFISBA) | European Commission/Joint Research Center (https://effis.jrc.ec.europa.eu/applications/data-and-services, accessed on 1 September 2022) | $62.5 \times 103\ m^2$ |

## 2. Materials and Methods

### 2.1. Land Use/Land Cover Changes Dataset

Information on LULCCs came from the CORINE land cover (CLC) database [31,32] provided by the European Environment Agency (EEA) [33,34]. LULCC data for the three periods (CHA 2000–2006, CHA 2006–2012 and 2012–2018) can be easily retrieved from the Copernicus CLC website (https://land.copernicus.eu/pan-european/corine-land-cover, accessed on 1 September 2022). Users can browse to find/select products, and each product's webpage contains a Map view, Metadata and a Download Table. The CORINE programme was initiated in the European Union in 1985 and is currently coordinated by the EEA in the framework of the EU Copernicus programme to produce land cover maps for the whole of Europe. Map updates were made available in 2000, 2006, 2012 and 2018 (with a minimum time consistency of $\pm 1$ year), allowing for multitemporal analysis of LULCC. The CLC minimum cartographic unit is 25 ha (Table 1), with a minimum geometric accuracy of 100 m and a thematic accuracy of over 85% [35]. A detailed description of the CLC can be found in the technical guides [36] and guidelines [37], user manual [34], including the nomenclature [38], and validation studies e.g., [39].

Time series of CLC were complemented by change layers (hereafter named CLCC), highlighting changes in land cover across Europe (Figure 1). The CLCC is not affected by changes in the methodology affecting two consecutive CLC datasets, because it is based on the revised version of the CLC dataset of the before-layer year, which is derived employing the same methodology as the after layer [40]. Additionally, in the past, CLCC has revealed irrelevant inconsistencies in comparison to CLC [41].

In the present study, we considered the three last CLCC datasets, which correspond to the differences between two consecutive inventories and reflect the changes that occurred in each of the CLCC periods: 2000–2006, 2006–2012 and 2012–2018. Finally, it should be mentioned that the percentage of changes over the total area concerning the starting year in each period corresponded to a small fraction (less than 2%) of the CLCC total area and mainly occurred (40%) in the 2006–2012 period (Table 2).

### 2.2. European Burnt Area Dataset

In this study, the Burnt Areas product [42] of the European Forest Fire Information System (EFFISBA) was considered the official European BA dataset (EFFISBA) for the entire 2000–2019 study period (Figure 2). The EFFISBA data used in this study were requested using the Data Request Form (available at https://effis.jrc.ec.europa.eu/apps/data.request.form/, accessed on 1 September 2022). This dataset is described in detail in [42]. EFFIS maps the BA based on two daily 250 m MODIS sensor images of the NASA TERRA and AQUA satellites using a semi-automatic procedure [43], the active-fire detection product and the news application. The spatial resolution of MODIS allows for accurately mapping fire with BA > 40 ha, but recent remote-sensing-based burnt-scar mapping capabilities allow for consistently mapping areas larger than 10 ha, which corresponds to mapping nearly 90% of the largest BA over Europe. The EFFISBA has been continuously validated since 1998, especially in the countries of the Mediterranean basin most affected by wildfires [44,45].

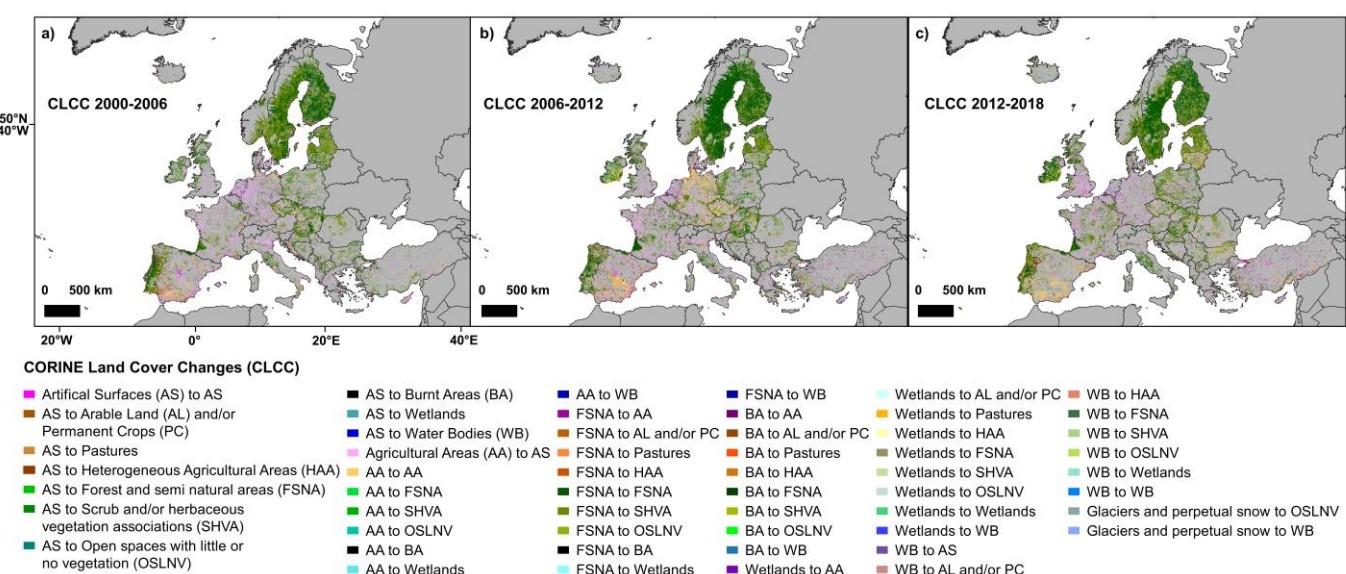

**Figure 1.** CORINE land cover change (CLCC) inventory for the (**a**) 2000–2006, (**b**) 2006–2012 and (**c**) 2012–2018 periods.

**Table 2.** The total area of each CORINE Land Cover Change (CLCC) inventory in absolute (TCLCC, ha) and relative terms (TCLCC/ΣCLCC, %) for the 2000–2006, 2006–2012 and 2012–2018 periods.

| Area | CLCC 2000–2006 | CLCC 2006–2012 | CLCC 2012–2018 | ΣCLCC 2000–2018 |
|---|---|---|---|---|
| TCLCC (ha) | 7,162,902.2 | 9,374,694.2 | 7,305,670.1 | 23,843,266.5 |
| TCLCC/ΣCLCC (%) | 30% | 39% | 31% | 100% |

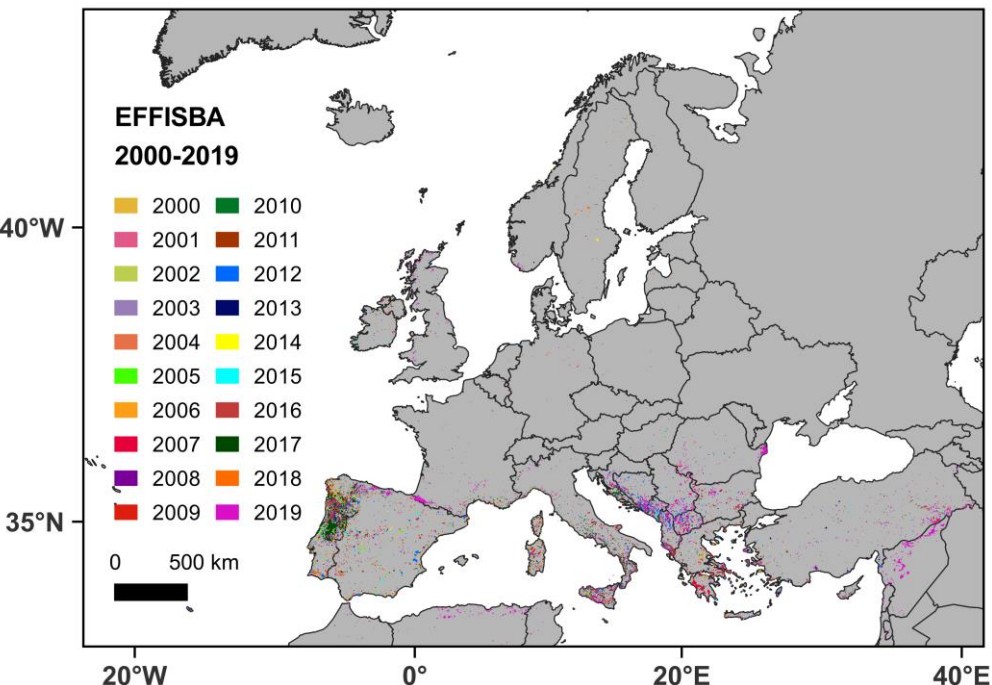

**Figure 2.** Burnt area polygons in Europe for the 2000–2019 period considering the European Forest Fire Information System (EFFISBA). The EFFISBA dataset includes polygons of burnt areas outside Europe, which were excluded from the analysis.

Other BA datasets are available based on satellite imagery, such as GlobFire [46], ESA FireCCI (https://climate.esa.int/en/projects/fire/, accessed on 30 September 2022) and Global Fire Atlas (https://www.globalfiredata.org/fireatlas.html, accessed on 30 September 2022) datasets; however, some of these datasets do not include small wildfire events, and, in other cases, they only cover a shorter period. Finally, we decided to use the EFFISBA dataset because it includes data for the entire study period (2000–2019) and is considered a reference BA dataset at the European level.

Based on this BA dataset, the five countries analyzed in the present study, namely Portugal, Spain, Greece, Italy and France, account for 73% of the total number of wildfires and 81% of the total BA over the entirety of Europe. The temporal evolution of the annual burnt area in Europe and in each of the Southern European countries (Figure 3) illustrates that the countries most affected by fires are Portugal and Spain, although, in some years, Greece and Italy also recorded high values of burnt area, and they reveal high interannual variability in the burnt area in all these spatial domains.

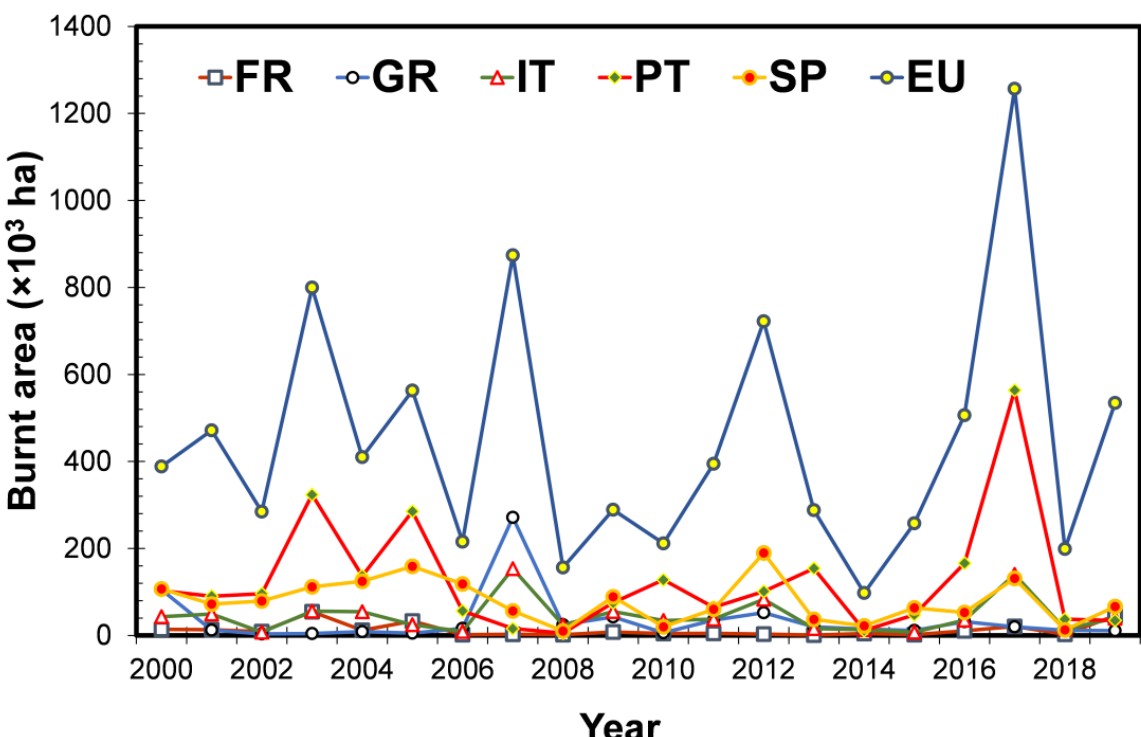

**Figure 3.** Annual burnt area in Europe and each of the Southern European countries for the 2000–2019 period considering the European Forest Fire Information System (EFFISBA).

### 2.3. Indicators to Assess Land Use/Land Cover Changes

We defined different indicators to assess the relationship between LULCCs and BA, as well as to assess the transitions between the LULC area and overlapping burnt areas. These are:

- CLCC Lost Area (CLCCLA), defined as the area lost in each CLC class between two consecutive CLC inventories.
- CLCC Gained Area (CLCCGA), defined as the area gained in each CLC class between two consecutive CLC inventories.
- Relative Total Lost Area (RTLA), defined as the CLCCLA evaluated for each period and each CLC class, divided by the total CLCC area (TCLCC) for the same period:

$$\text{RTLA}(\%) = \frac{\text{CLCCLA}}{\text{TCLCC}} \times 100 \tag{1}$$

- Relative Total Gained Area (RTGA), defined as the CLCCGA evaluated for each period and each CLC class, divided by the total CLCC area (TCLCC) for the same period:

$$\text{RTGA}(\%) = \frac{\text{CLCCGA}}{\text{TCLCC}} \times 100 \tag{2}$$

- Net Changed Area (NCA), defined for each CLC class as the difference between CLCCGA and CLCCLA:

$$\text{NCA} = \text{CLCCGA} - \text{CLCCLA} \tag{3}$$

These indicators were computed for the entirety of Europe and each of the five southern European countries separately.

### 2.4. Land Use/Land Cover Change Characterization within Burnt Areas

The quantitative assessment of the LULCCs within BAs allows us to identify the CLC classes more and less prone to wildfire occurrence. To this end, we developed an indicator allowing us to estimate, for each CLCC class and each period, the portion of CLCC area (ACLCC) coincident with the overlapping burnt area (EFFISBA), all divided by the total area of the corresponding CLCC inventory (TCLCC):

$$\text{CLCCEFFISBA}(\%) = \frac{\text{ACLCC} \cap \text{EFFISBA for each CLCC class}}{\text{TCLCC}} \times 100 \tag{4}$$

To guarantee that the two inventories—CLCC and EFFISBA—cover the same frame period, the yearly EFFISBAs were aggregated into overlapping frame periods of 5 years, corresponding to the three CLCC periods $\pm$ 1 year (to account for the CLC time consistency), which results in the following: 2000–2007, 2006–2013 and 2012–2019.

### 3. Results

### 3.1. Quantitative Assessment of Land Use/Land Cover Changes in Europe

The relative total lost area (RTLA) and relative total gained area (RTGA) results presented here are focused only on the five LULC classes that underwent the most significant changes (Figure 4). About three-quarters of the total area affected by land cover changes (TCLCCs) in Europe were forests and shrubs and/or herbaceous vegetation associations (hereafter, shrubs). The average RTLA for the three CLCC periods (2000–2006, 2006–2012 and 2012–2018) for these two classes was 44% and 32%, respectively (Figure 4a). A similar pattern was observed in the five investigated Southern European countries, especially in Portugal and France, which present an RTLA of 49% and 51% for forests and 30% and 26% for shrubs, respectively. In general, for all five South Mediterranean countries and the entirety of Europe, forests and shrubs experienced significant changes in all the CLCC periods, followed by permanent crops and, at a greater distance, arable lands.

Temporal trends are also worth mentioning. In Europe, from the first (2000–2006) to the second (2006–2012) and then to the third (2012–2018) CLCC period, the RTLA in forests started decreasing and then increased, while the RTLA in shrubs showed an opposite trend. Similar RTLA trends can also be observed in Portugal and, somehow, in France.

The results for RTGA (Figure 4b) are similar, with the difference that, in Europe, the value for shrubs is higher (46%) than for forests (28%). The RTGA in Southern European countries is also higher for these two CLC classes, especially in Portugal and France, where RTGA for shrubs (54% and 51%, respectively) and forests (23% in both countries) accounts for most of the gained area. In Spain, the CLC class with the highest RTGA is also shrubs (34%), followed by agricultural areas (permanent crops 16% and arable land 12%).

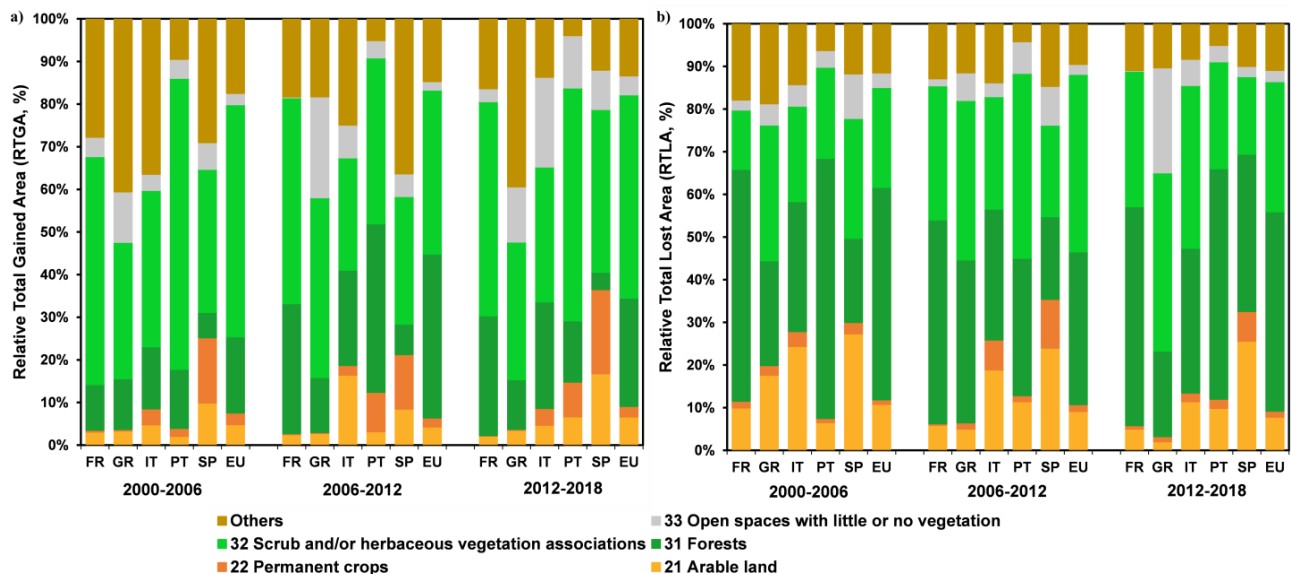

**Figure 4.** Relative total lost area (RTLA, panel **a**) and relative total gained area (RTGA, panel **b**) based on CORINE land cover changes (CLCCs) for Europe (EU), France (FR), Greece (GR), Italy (IT), Portugal (PT) and Spain (SP), in the CLCC periods: 2000–2006, 2006–2012 and 2012–2018.

The results of the NCA (Figure 5) show high temporal variability in Europe, with the highest values in the first CLCC period (+2223 ha for shrubs and −2286 ha for forests), high values in the last (+1258 ha for shrubs and −1564 ha for forests) and the lowest values in the central period (+44 ha for permanent crops and–29 ha for open spaces with little or no vegetation). The five Southern European countries tend to follow the same global trend. The top countries in terms of NCA were Portugal, Spain and France, in this order, except during the central CLCC period, when Portugal presented the third-lowest NCA.

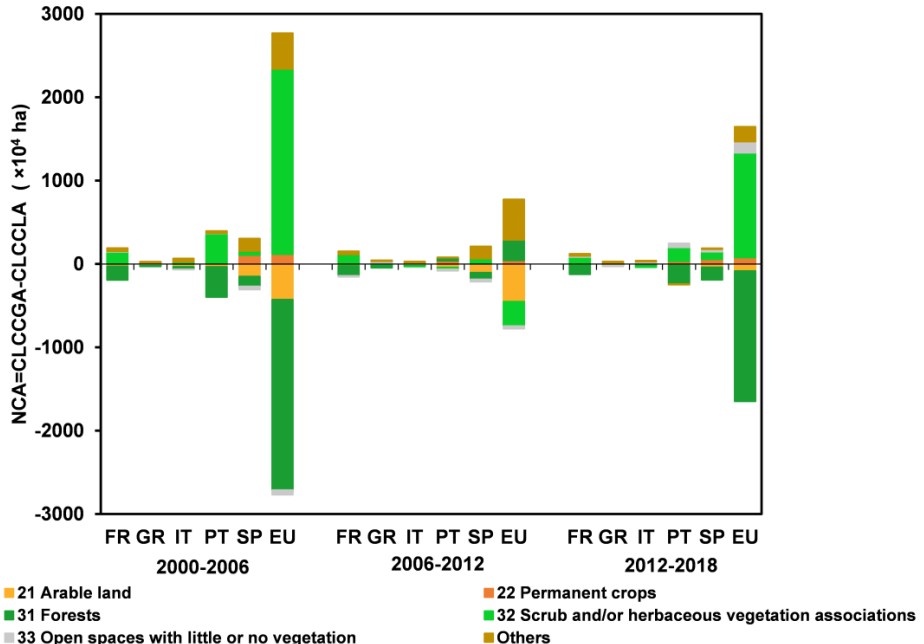

**Figure 5.** Net Changed Area (NCA), computed as the difference between the CORINE land cover changes (CLCC) Gained Area (CLCCGA) and the CLCC Lost Area (CLCCLA) for the CLC layers: 21—arable land; 22—permanent crops; 31—forests; 32—scrub and/or herbaceous vegetation associations; 33—open spaces with little or no vegetation; and others; for the spatial domains (France, Fr, Greece, GR, Italy, IT, Portugal, PT, Spain, SP and Europe, EU) and CLCC periods (2000–2006, 2006–2012 and 2012–2018) on the x-axis.

The CLC classes with higher and lower NCA are not uniformly distributed. However, generally, the results are similar for Europe and each of the five southern countries in the three CLCC periods. The features that stand out are the negative values of the NCA for forests and arable lands and the positive values for shrubs. These are, in general, the CLC classes with the highest absolute value of NCA, particularly for forests and shrubs in Europe, Portugal and France. This result suggests the potential effect of wildfires on LULCCs by converting forest areas into shrublands.

### 3.2. Quantitative Assessment of Land Use/Land Cover Changes within Burnt Areas

The analysis described in this sub-section is focused only on the top LULCC that occurred within BAs considering the European Forest Fire Information System Burnt Area dataset (EFFISBA) for each CLCC period across Europe and in each Southern European country (Figure 6).

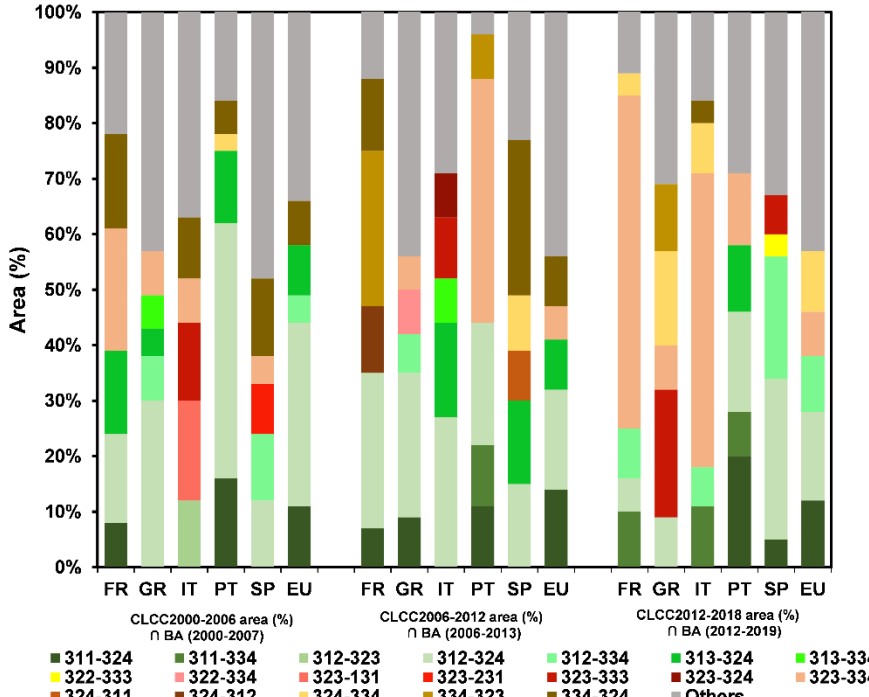

**Figure 6.** Top CORINE Land Cover Change (CLCC) class area (%) when intersecting European Forest Fire Information System Burnt Area dataset (EFFISBA) with each CLCC period (2000–2006, 2006–2012 and 2012–2018) for each study area: Europe (EU), France (FR), Greece (GR), Italy (IT), Portugal (PT) and Spain (SP). It should also be read: mineral extraction sites (CLC131), pastures (231), broad-leaved forest (CLC311), coniferous forest (CLC312), mixed forest (CLC313), moors and heathland (CLC322), sclerophyllous vegetation (CLC323) and transitional woodland–shrub (CLC324), sparsely vegetated areas (CLC333) and burnt area (CLC334).

As for the previous analyses, high variability and similarity in the pattern's distribution of the different transitions have been revealed. It is essential to underline that the CLCC includes information about the land cover before and after the change. The classes of origin of the transitions with the highest relative area within an area affected by a forest fire (CLCCEFFISBA) are forests and shrubs, namely: coniferous forests (CLC312), broad-leaved forests (CLC311), mixed forests (CLC313), sclerophyllous vegetation (CLC323) and transitional woodland–shrub (CLC324).

The average CLCCEFFISBA (for all the frame periods) for forests is 42% and for the abovementioned sub-classes is 24% (CLC312), 12% (CLC311) and 9% (CLC313), while for shrub classes (CLC323 and CLC324) is much lower (4%). Over the three frame periods, CLCCEFFISBA shows a decreasing trend, mainly due to the negative trend observed in

coniferous forests (CLC312). The results for the five southern countries differ from those obtained for the entirety of Europe, but, in general, they follow the same patterns. The average CLCCEFFISBA (for all the periods) for forests ranges from 24% in Italy to 47% in Greece and 50% in Portugal. The decreasing trend of CLCCEFFISBA for coniferous forests (CLC312) in Portugal is particularly significant (from 46% to 20%) and was not observed in the other Southern European countries.

The transitions with higher CLCCEFFISBA in Europe are from coniferous forest (CLC312) to transitional woodland–shrub (CLC324) and from broad-leaved forest (CLC311) to transitional woodland–shrub (CLC324). This result was observed in all the periods but with different values: respectively, 33% and 11% in the first period, 18% and 14% in the second period and 16% and 12% in the last period. The third (mixed forest, CLC313, to transitional woodland–shrub, CLC324) and fourth (burnt area, CLC334, to transitional woodland–shrub, CLC324) main transitions with the highest CLCCEFFISBA are also the same in the first and second periods but become the second and third most important transitions in the third period. However, the five transitions with the highest CLCCEFFISBA for any Southern European country vary with time.

In Europe, the largest CLCC areas exposed to wildfires were primarily found in the transitions to transitional woodland–shrub (CLC324) and burnt areas (CLC334). The CLCC areas intersecting EFFISBA for these two transitions present opposite sign trends. The intersecting area for transitions to transitional woodland–shrub (CLC324) decreases from 62%, in the first period, to 28%, in the last period, while the transitions to burnt area (CLC334) increase from 5% to 29% in the same period. Southern European countries present trends of equal sign, except for Spain, where the intersecting area increases for the transitions to transitional woodland–shrub (26% to 60%) and decreases for the transitions to burnt area (17% to 7%). Trends for other Southern European countries are quite different. Portugal is the most uncommon case, where the intersecting area for the transitions to transitional woodland–shrub decreased from 80% to 7%, but transitions to burnt area increased from 3% to 64%. In France, the corresponding results are 56% to 6% and 22% to 82%. For Greece and Italy, these trends are not always so impressive.

## 4. Discussion

There is a widespread understanding that fires and LULCCs are interrelated, but it is important to evolve from answers to the questions of how and why to the questions of what, when and where. This is the reason why this study aimed to answer the research questions identified in the Introduction, performing a quantitative assessment of the relationship between fires and LULCCs in Europe. In general, the quantitative assessment usually consists of the systematic empirical investigation of phenomena, using statistics and mathematics analysis of large numerical (quantitative) datasets, with results that are also numerical, which facilitate the highlighting of changes and differences [47]. Quantitative assessment is, therefore, suitable for analyzing large databases of CLC, CLCC and BA and to quantify and compare results of the LULCC assessment in general and specifically within the BA, over time across the European continent and in specific countries. The advantages of the quantitative assessment include, but are not limited to, the following: (a) the result is numerical, representing research and, therefore, often considered to be objectively fact based, measurable and observable facts; (b) allows for easier comparison of data/results; and (c) based on quantitative valuation indicators, which, in our case, were the six developed different indicators, mainly based on relative measurements that are necessary for the adequate comparison and understanding of absolute measurement [48]. In the following subsections, we will discuss the results of the quantitative assessment considering the indicators explained in our methodology.

### 4.1. Justification and Validation of the Databases Used

In this subsection, we discuss the rationale for selecting the databases used for this study. A large number of studies existing in the literature compared CLC with other global LULC databases for various purposes [49,50]. Although there are some discrepancies on a local scale, CLC is considered a reliable source of information and a reference LULC dataset at the European level. For example, several studies showed good accuracy and consistency on the regional scale of CLC [40,51]. In a recent study, Reinhart et al. [49] compared CLC with ESA climate change initiative land cover (ESA CCI) over Eastern Europe and the Baltic States and did not find any substantial difference between proportional area and majority comparison. Fonte et al. [52] found that, on the Portuguese scale, ESA CCI overestimated the forest/natural vegetation CCL class by 11–13%. Moreover, CLC has been broadly used across Europe [53], including land use change research [40,54,55] and wildfire research [51,56–58].

### 4.2. Quantitative Assessment of Land Use/Land Cover Changes in Europe

Analyses performed in Europe reveal that most of the LULCCs occurred in the classes of forests and shrubs (75%), followed by, in a much smaller proportion (only 5%), arable lands (Figure 3). As observed by other authors, we can infer that LULCCs mainly occurred in areas used for timber harvesting or other forestry products, in areas under agricultural rotation systems used for annually harvested plants and fallow lands [35] and in rural areas disturbed or degraded by various factors, including droughts and wildfires [14]. Wildfires require vegetable fuels [59] and usually burn the forest, scrub and agricultural areas [30,60]. On the other hand, after a wildfire, the affected areas are generally occupied by new agroforestry plantations or by natural/indigenous vegetation [61].

Specific results for the relative total lost area (RTLA) and the relative total gained area (RTGA) (Figure 4) for the five Southern European countries follow this global trend, although the proportions of the CORINE land cover (CLC) classes vary among the countries, possibly due to different characteristics of climate, LULC management, urban and rural development and economic policies. For example, the increase in Portugal of forest and seminatural areas and inland waters during the first CLCC period is possibly related to the increase in the wildland and urban interface [30] and to the construction of at least ten dams between 2000 and 2006, including Alqueva, which is Western Europe's largest artificial lake, with an area of 250 km$^2$ (fully filled in 2010) [62]. Additionally, the increase in dense forests in Greece might be related to the stopping of resin tapping, grazing and other traditional forest uses [63]. Additionally, the RTLA and RTGA of arable land in Italy might be related to the industrialization of agriculture, socioeconomic change and the globalization of agricultural markets [64].

The open space with little or no vegetation (CLC 33 class) is one of the five level-2 classes that most changed in the study period, especially in a small number of cases (Figure 3). The CLC33 class includes the burnt area (CLC334) and the sparsely vegetated areas (CLC333), which tend to reflect the impacts of wildfires and climate variability and change, including drought and other causes of desertification worldwide [65,66] and in Southern Europe [67–69].

If the results of RTLA and RTGA support the evidence of the relationship between wildfires and LULCCs, the results about the net changed area (NCA) (Figure 5) reinforce this hypothesis even more. In general, NCA for Europe as a whole and most of the southern investigated countries decreased from the first to the second CLCC periods, but it increased in the recent CLCC period. Additionally, in most of the five Southern European countries, the NCA presents a net loss in forests and arable Land and a net gain in shrubs, which suggests a potential fire effect in LULCCs. Of course, not all the changes in NCA are due to wildfires. In Spain, the decrease in arable land is essentially due to transitions to artificial surfaces and other types of agriculture, while the increase in the class "Others" is due to the increase in artificial surfaces (mainly mine, dump and construction sites), heterogeneous agriculture areas and pastures.

*4.3. Quantitative Assessment of Land Use/Land Cover Changes within Burnt Areas*

The analysis of LULC transitions within the BAs was performed considering the overlap between CLCC and EFFISBA (Figure 6) for each period and each Southern European country. The results of this analysis indicate that most of the BAs in Europe occurred in forests and gave origin to transitional woodland/shrub or again to burnt areas. Local analysis has shown similar trends of forests or shrubs to transitional woodland–shrub or burnt areas, apart from Italy, which presents an LULCC between sclerophyllous vegetation and mineral extraction sites. It is essential to mention that CLC class transitional woodland–shrub is defined as "Transitional bushy and herbaceous vegetation with occasional scattered trees", which "can represent woodland degradation, forest regeneration/recolonization or natural succession". At the same time, the transition from BA to BA indicates a high wildfire recurrence. Therefore, these results may reflect the possible success or failure of environmental system recovery, strategies and actions of fire management and prevention in different countries. These results are in line with previous studies. For example, Bajocco et al. [70] identified LULCCs associated with high fire occurrence in Sardinia (Italy), namely agricultural areas to shrubs, burnt areas or sparsely vegetated areas, which have been associated with the progressive abandonment of cultivated land. Oliveira et al. [71] showed that wildfires in Southern European countries mainly occur in shrubs and grasslands. Badia et al. [72] studied the fire vulnerability of the wildland and urban interface according to LULCCs and found that scrub vegetation in BAs has been succeeded by dense woods and burnt areas, and most of the wildfires in mountain agro–silvopastoral areas were in densely wooded areas, in the scrub and farm fields. Previous studies have shown that most of the BAs occurred in forests that were converted to degraded native vegetation or agricultural lands [61].

## 5. Conclusions

The immediate consequence of wildfires is the partial or total burning of vegetation fuels, including forested areas, grass and alpine/tundra vegetation. Thus, this hazard primarily affects natural resources but can also have an impact on rural/semi-natural areas. As a result, wildfires led to land use/land cover changes and vice versa, i.e., land use/land cover changes may influence the fire regime and their incidence.

Although the relationship between wildfires and land use/land cover changes seems to be evident, a rigorous analysis of the interdependency between these two factors at the European scale needs to be better investigated to improve landscape and fire management. This was the prime motivation for the present study. To this aim, we defined different indicators, allowing us to (i) assess the main land use/land cover changes that occurred in Europe between 2000 and 2019, and (ii) quantify and better understand the relationship between these changes and wildfires.

The land use/land cover classes that experienced the greatest changes across Europe in the three reference periods (2000–2006, 2006–2012 and 2012–2018) were forests, with an average of 44% of total changes, and shrubs and/or herbaceous vegetation associations, with an average of 32%. As a general trend, within the five countries most prone to fires (i.e., Portugal, Spain, France, Italy and Greece), we observed a decrease in forests and arable land and an increase in shrubs, which suggests the impact of wildfires in shaping the natural and anthropogenic environment. This assumption was better evaluated and confirmed by a deeper analysis, performed both at the European and national levels. The results show that most of the burnt areas (BAs) occurred in forests (42% for entirety of Europe), with a predominance in coniferous forests, and that the transitions from BA generally were to transitional woodland/shrub or again to burnt areas. The results at the national level follow the same trend. The first transition can be partially due to the regeneration/recolonization of the vegetation after a wildfire event, while the transition from BA to BA indicates a high wildfire recurrence. These findings are in line with previous studies and provide a deeper understanding of the process at the global level, paving the way for further analyses of fire intensity and frequency with coupled environmental elements of land cover and climate

changes. Finally, we believe that our study contributes to improving the ecology of the environmental system and fire management in Europe and may even motivate similar quantitative assessments in other regions of the world.

**Author Contributions:** Conceptualization, M.P. and J.P.; methodology, M.P. and J.P.; software, J.P.; validation, M.P., J.P., M.T., Z.S. and N.K.; formal analysis, J.P.; investigation, M.P., J.P., M.T., Z.S. and N.K.; resources, M.P. and J.P.; data curation, J.P.; writing—original draft preparation, J.P., M.P. and M.T.; writing—review and editing, M.P., J.P., M.T., Z.S. and N.K.; visualization, J.P.; supervision, M.P., M.T. and N.K.; project administration, M.P.; funding acquisition, M.P. All authors have read and agreed to the published version of the manuscript.

**Funding:** This work was financed by the project FRISCO—managing Fire-induced RISks of water quality Contamination (FCT ref.: PCIF/MPG/0044/2018). The study was also supported by funding attributed to the CE3C research centre (UIDB/00329/2020) and National Funds through FCT—Foundation for Science and Technology—under the project UIDB/04033/2020.

**Data Availability Statement:** All data used in this study are freely accessible on the platforms of data providers, referred to in Section 2. The datasets generated and/or analyzed during the current study are available from the corresponding author upon reasonable request.

**Acknowledgments:** We wish to thank the Copernicus Land Monitoring Service for providing the CORINE Land Cover data and acknowledge that the Copernicus Land Monitoring Service products and services are produced with funding by the European Union. Burnt area data were provided by the European Forest Fire Information System—EFFIS (https://effis.jrc.ec.europa.eu, accessed on 1 September 2022) of the European Commission Joint Research Centre.

**Conflicts of Interest:** The authors declare no conflict of interest. The funders had no role in the design of the study; in the collection, analyses, or interpretation of data; in the writing of the manuscript; or in the decision to publish the results.

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
