# Peer review of "Quantitative Assessment of the Relationship between Land Use/Land Cover Changes and Wildfires in Southern Europe"

_fire, doi:10.3390/fire6050198_

Round 1
Reviewer 1 Report
Dear authors,
Your manuscript presents an interesting work analysing the relationship between LULCC and wildfires. This is a known topic that has been done at local or regional level, but your study is at European level and in the five most affected countries by fire. The datasets you have used are available and the study can be replicated, as you elaborated quantitative indicators, obtaining interesting trends and results that can be compared with previous studies. Nice the chosen time periods to avoid temporal discrepancies.
I found the work interesting and acceptable to be published after some revision. My main concerns are related to some parts of the text or figures that need to be clarified. And why you have considered you have found a good match between burned area data sources (more explanation is needed). Please find below specific comments by section and by line, including figures or tables.
Specific comments:
Summary
Lines 23-24: The main results are the following: (i) LULCC analysis reveals a net loss in Forests and Arable Land (…)à Past tense when presenting the results, as it is a study already finished.
Introduction
Line 35: Shrubs or scrubs?
Lines 55-56: Climate change can play a role by leading to changes in land use and management practices [31] with extreme weather events, such as droughts and heat waves, impacting vegetation and increasing fire risk [31–33]. à these references are referred to Portugal, any other at global scale as it is suggested by the sentence?
Lines 57-58: LULCC was found to increase the fire occurrence probability in the vicinity of urban areas and human infrastructures, also because of the anthropogenic origin of most wildfires [34,35]. àwhere? Globally? Europe? Please clarify
Materials and methods
Line 154: 2.3. National burnt area dataset à how is this dataset calculated? Linked to section 3.3. Comparison between national and global burnt area dataset: the case study of Portugal
Line 179: CLCC Lost Area (CLCCLA), is defined as the area lost in each CLC class. à from one period to another?
Results
3.1. Quantitative assessment of land use/land cover changes in Europe
Lines 214-215: The relative total lost area (RTLA) and relative total gained area (RTGA) results presented here are focused only on the five LULC classes that had undergone the most significant changes à over x% ??
Line 232: Temporal trends are also worth mentioning. In Europe, from the first to the second and then to the third CLCC periodà I would remind the periods
3.2. Quantitative assessment of land use/land cover changes within burnt areas
Line 287: The third (Mixed forest, CLC313, to Transitional woodland-shrub, CLC324) and fourth (Burnt Area, CLC334, to Transitional woodland-shrub, CLC324) main transitions with the highest CLCCEFFISBA are also the same in the first and second periods. àand what happened in the third period?
Line 290: Figure 4. In the caption not all classes are being described i.e. CLC231
This figure it is sometimes a bit difficult to be read. I would suggest to choose the same colour scale for the transitions to the same category, i.e. from X to 324, green scale (or any other)
3.3. Comparison between national and global burnt area dataset: the case study of Portugal
I would like to know how the Portuguese BA dataset is calculated in section 2.3
Discussion
Line 356: possibly due to different characteristics of climate, LULC management, urban and rural development, and economic policiesàany global reference here?
Line 413: number of fires in Europe, the vast majority of which are small in size [73] and can escape differently from the different detection and mapping methodologiesà for that reason it would be interesting to know or to write a specific reference explaining a bit more the Portuguese BA dataset.
Conclusions
Line 447: As a result, wildfires led to land use/land cover changes and vice-versa, .e.--> typo, missing the i
Line 475: Results revealed a good match between the official European wildfires dataset (EFFISBA) and the national Portuguese BA datasetà how do you justify the match is good enough? Please revise
Author Response
Answers to Reviewer 1
Dear authors,
Your manuscript presents an interesting work analysing the relationship between LULCC and wildfires. This is a known topic that has been done at local or regional level, but your study is at European level and in the five most affected countries by fire. The datasets you have used are available and the study can be replicated, as you elaborated quantitative indicators, obtaining interesting trends and results that can be compared with previous studies. Nice the chosen time periods to avoid temporal discrepancies.
I found the work interesting and acceptable to be published after some revision. My main concerns are related to some parts of the text or figures that need to be clarified. And why you have considered you have found a good match between burned area data sources (more explanation is needed). Please find below specific comments by section and by line, including figures or tables.
Answer: We are very grateful to the reviewer for the time and effort taken to revise our manuscript as well as for the suggestions for changes that we are sure will improve the quality of our manuscript.
Specific comments:
Summary
Lines 23-24: The main results are the following: (i) LULCC analysis reveals a net loss in Forests and Arable Land (…)à Past tense when presenting the results, as it is a study already finished.
Answer: We regret the typo and correct the verb accordingly.
Introduction
Line 35: Shrubs or scrubs?
Answer: Our idea was to refer to the CLC class we defined as Scrubs. However, we did not define this class in the abstract. We believe that it is not necessary to do it in the abstract and correct the word.
Lines 55-56: Climate change can play a role by leading to changes in land use and management practices [31] with extreme weather events, such as droughts and heat waves, impacting vegetation and increasing fire risk [31–33]. à these references are referred to Portugal, any other at global scale as it is suggested by the sentence?
Answer: Study 31 was performed on a global scale; studies 32 and 33 were formed in Portugal. The role of droughts and heat waves on vegetation state and fire risk is well known in many locations. However, we understand the question of the reviewer and we added a few more relevant citations of studies about these topics in the world and in Europe.
Lines 57-58: LULCC was found to increase the fire occurrence probability in the vicinity of urban areas and human infrastructures, also because of the anthropogenic origin of most wildfires [34,35]. àwhere? Globally? Europe? Please clarify
Answer: We clarify where LULCC increase fire probability.
Materials and methods
Line 154: 2.3. National burnt area dataset à how is this dataset calculated? Linked to section 3.3. Comparison between national and global burnt area dataset: the case study of Portugal
Answer: Another reviewer suggested removing from the manuscript all parts of the second objective of the initial work, in which a special case study from Portugal was investigated. We agreed with this suggestion and, consequently, all the text related to this part was removed.
Line 179: CLCC Lost Area (CLCCLA), is defined as the area lost in each CLC class. à from one period to another?
Answer: In the new version of the manuscript, we clarified how CLCCLA and CLCCGA have been computed.
Results
3.1. Quantitative assessment of land use/land cover changes in Europe
Lines 214-215: The relative total lost area (RTLA) and relative total gained area (RTGA) results presented here are focused only on the five LULC classes that had undergone the most significant changes à over x% ??
Answer: We thank the reviewer for the opportunity to clarify this issue. We don't set a limit as the RTLA and RTGA for each class vary greatly. For RTGA, class 32 ranges from 30% to 70%, class 21 ranges from 2% to 20%, class 22 ranges from 0 to 20%, class 31 ranges from 4 to 30%, and class 33 varies between 0 and 24%. For all other classes, the RTGA was always less than 10%. For RTLA, class 21 ranges from 2 to 30%, class 22 ranges from 0 to 11%, class 31 ranges from 20 to 61%, class 32 ranges from 14 to 45%, class 33 ranges from 0 and 25%, and the others range from 0 to 5%. So, instead of defining a limit, we decided to represent the RTLA and RTGA for the 5 classes that presented, on average, the highest values. This information, at least for the most part, is already provided to the reader, in the sense that the heights of the bars in the old Figures 3 and 4, allow checking the range of values of each class and perceiving in each case that the classes not represented contribute very small amounts to RTLA and RTGA.
Line 232: Temporal trends are also worth mentioning. In Europe, from the first to the second and then to the third CLCC periodà I would remind the periods
Answer: We reminded the periods.
3.2. Quantitative assessment of land use/land cover changes within burnt areas
Line 287: The third (Mixed forest, CLC313, to Transitional woodland-shrub, CLC324) and fourth (Burnt Area, CLC334, to Transitional woodland-shrub, CLC324) main transitions with the highest CLCCEFFISBA are also the same in the first and second periods. àand what happened in the third period?
Answer: We included the relative positions of these transitions in the third period.
Line 290: Figure 4. In the caption, not all classes are being described i.e. CLC231
Answer: We thank the reviewer for spotting the flaw and we have added the description of class CLC231 (Pastures).
This figure it is sometimes a bit difficult to be read. I would suggest to choose the same colour scale for the transitions to the same category, i.e. from X to 324, green scale (or any other)
Answer: We agree with the reviewer that this figure is a little difficult to read. The main reason is the high number of transitions and the lack of a small set of dominant transitions. To facilitate the interpretation of the graphical representation we used a qualitative color scale and a criterion (as suggested by the reviewer) of using similar colors for transitions of classes of the same type. For example, all forest class transitions (31) were represented by shades of green; transitions from classes of Scrub and/or herbaceous vegetation associations (32) were represented with yellow-red tones, and from classes of Open spaces with little or no vegetation (33) in shades of dry green-brown. Specifically, the reviewer's suggestion is precisely the opposite of ours: choose colours, not according to the class of origin, but according to the class of arrival. We are convinced that the two suggestions will be approximately equivalent, with regard to the difficult task of making the figure more easily readable.
3.3. Comparison between national and global burnt area dataset: the case study of Portugal
I would like to know how the Portuguese BA dataset is calculated in section 2.3
Answer: As mentioned before, in the new version of the present manuscript we removed all parts concerning this second objective, related to the Portuguese BA dataset.
Discussion
Line 356: possibly due to different characteristics of climate, LULC management, urban and rural development, and economic policiesàany global reference here?
Answer: In this case, we mentioned specifically the five southern European countries, as explained in the previous lines “…Specific results for the relative total lost area (RTLA) and the relative total gained area (RTGA) (Figure 3) for the five southern European countries follow this global trend, although the proportions of the CORINE land cover (CLC) classes vary among the countries, possibly due to different characteristics of climate, LULC management, urban and rural development, and economic policies.”
Line 413: number of fires in Europe, the vast majority of which are small in size [73] and can escape differently from the different detection and mapping methodologiesà for that reason it would be interesting to know or to write a specific reference explaining a bit more the Portuguese BA dataset.
Answer: As mentioned before all parts of the second objective of the initial work were removed from the manuscript, including the entire 4.3 section.
Conclusions
Line 447: As a result, wildfires led to land use/land cover changes and vice-versa, .e.--> typo, missing the i
Answer: We thank the reviewer for the opportunity to correct the typo.
Line 475: Results revealed a good match between the official European wildfires dataset (EFFISBA) and the national Portuguese BA datasetà how do you justify the match is good enough? Please revise
Answer: As mentioned before all parts of the second objective of the initial work were removed from the manuscript, including this part.
Reviewer 2 Report
The article is interesting and useful, thank you very much. There are no comments, I would like to shorten the introduction a little, remove sentences that are not related to the main idea of the article. For example, lines 49-54.
Author Response
Answers to Reviewer 2
The article is interesting and useful, thank you very much. There are no comments, I would like to shorten the introduction a little, remove sentences that are not related to the main idea of the article. For example, lines 49-54.
Answer: We are very grateful to the reviewer for the time and effort taken to revise our manuscript as well as for the suggestions for changes that we are sure will improve the quality of our manuscript.
As suggested by the reviewer, we remove the text in lines 49-54.
Reviewer 3 Report
The Reviewer's comments are in the attached PDF file.

Minor editing of English language required.
Author Response
Answers to Reviewer 3
Review on the manuscript MDPI fire-2368369
“Quantitative assessment of the relationship between land use/land cover changes and wildfires in Europe” by authors: Joana Parente, Marj Tonini, Zoi Stamou, Nikos Koutsias, Mario Pereira
Major Comments and Suggestions for the Authors
The authors of this manuscript investigated the relationship between land use/land cover (LULC)changes and wildfires. A methodology has been developed based on various indicators that allow us to quantify and better understand the transitions between LULC classes and burnt area (BA) in Europe over the past two decades (2000 – 2020). The author's results confirm the existence of a close relationship between wildfires and LULC changes in Europe.
Unfortunately, the design and methodology of the manuscript require corrections. The mandatory request for publication is removed from manuscript the part inclusive a second aim of work, in which is investigated a special case study of Portugal. At the same time, according to the reviewer, the results presented in this work are relevant and may be of interest to readers.
The MDPI fire-2368369 manuscript can be published after minor revisions.
Answer: We are very grateful to the reviewer for the time and effort taken to revise our manuscript as well as for the suggestions for changes that we are sure will improve the quality of our manuscript.
We agreed to remove from the manuscript all parts of the second objective of the initial work, in which a special case study from Portugal is investigated.
We also appreciate the reviewer's recommendation that our manuscript can be published after minor revisions.
Please see our answers to the reviewer’s major and minor comments below.
Major and Minor Comments:
Comment 1 (Title, keyword list)
The title of the manuscript and the list of keywords indicate that the object of this study is Europe. However, the Abstract says that changes in land use\land cover (LULC) are being investigated not only for the whole of Europe, but separately for the following countries: Portugal, Spain, France, Italy, and Greece. Further in the text of manuscript, more attention is paid to the study of changes in vegetation exactly in these five countries. It would probably be more correct to replace the word “Europe” with “Southern Europe” in the title of the manuscript and in the list of keywords.
Answer: We agree with the reviewer and changed the title and the list of keywords as suggested.
Comment 2 (lines 68–71)
“Additionally, we have as a secondary objective to evaluate possible discrepancies between BA datasets developed at the European and national scale (Table 1), for which we studied the case of Portugal. Our aim is not to formally compare BA databases, but only to assess similarities and differences that may justify the choice of databases for this type of study.”
Below is another comment about the object of the study.
The case study of Portugal as the second aim, highlighted in lines 68–71, contradicts what is written in the Title of the manuscript and in the Abstract. Thus, the question of investigating the connection between Portugal wildfires and land cover changes based on the ICNFBA dataset (Instituto da Conservação da Natureza e das Florestas Burnt Areas) may be the topic of another article, but this material can in no way be presented in this article. Therefore, if authors would like to write about the Portugal ICNFBA dataset, then for the correct description in this publication they need to additionally use the official national burnt area data sets provided by the forest services of Spain, Italy, France and Greece.
The Reviewer recommends deleting everything related to a special case study of Portugal.
Answer: We agree with the reviewer and, accordingly, removed everything related to a special case study of Portugal.
Comment 3 (Table 1)
The link to ICNFBA dataset is send reader to resource in the Portuguese language.
(https://icnf.pt/florestas/gfr/gfrgestaoinformacao/dfciinformacaocartografica)
Answer: Since we removed everything related to a special case study of Portugal, all references to ICNFBA were also removed.
Comment 4 (Table 1, line 39 and References [38] and [39])
38. EEA CORINE Land Cover 2020.
39. Büttner, G.; Kosztra, B.; Maucha, G.; Pataki, R.; Kleeschulte, S.; Hazeu, G.W.; Vittek, M.; Schroder, C.; Littkopf, A. Copernicus Land Monitoring Service - CORINE Land Cover. User Manual 2021.
For downloading CORINE land cover data, authors need to specify ftp or https. It also requires a real link to the PDF file that describes the CORINE Manual Guide.
Please correct the list of References.
Answer: We clarified how the data was obtained and corrected the list of references, as suggested.
Comment 5 (line 49-50)
“LULCC is both a cause and a consequence of global environmental changes [22]. Quantification of global land use changes revealed a human-dominated Earth system, driven by direct human actions as well as human-induced climate change [23].”
An increase in temperature by 0.5°C does not lead to the ignition of dry grass, shrubs or tree branches, therefore, the statement about the impact of climate change on the increase in the number of fires requires justification.
Note that the deliberate burning of large areas leads to a significant increase in wildfire aerosols, which, in turn, theoretically, according to anarchist climatologists, can lead to an increase in reflected solar radiation and cooling of the planet. The action of ground-based pyromaniac groups, as well as the analysis of the possibility of remote arson by using modern drones and satellite technologies, essentially in inaccessible or hard-to-reach mountainous areas, go far beyond the scope of this publication. Therefore, if the authors are not going to engage in catching pyromaniacs or study the problem of the formation of dry thunderstorms, it is recommended to avoid mentioning climate change, human-induced climate change and direct human actions.
Please make appropriate changes to the text of the manuscript.
Answer: We agree with the reviewer, in the sense that the manuscript is not about climate change, fire emissions, or causes. Another reviewer (reviewer 1) also suggested the removal of the text in lines 49-54 and we decide to change the manuscript accordingly
Comment 6 (Table 1, line 159, and References [54]).
Again, the manuscript lacks any background information about the European Forest Fire Information System Burned Area (EFFISBA). Please provide ftp or https links to download data about the EFFISBA burned area and the Manual Guide for EFFISBA description.
Answer: The EFFIS Burnt Area data used in this study was requested using the Data Request Form (available at https://effis.jrc.ec.europa.eu/apps/data.request.form/. In the new version of the manuscript, we clarify how EFFISBA data was obtained. We also clarify how the reader can obtain background information about EFFISBA.
Finally, line 159 of the manuscript is about the ICNFBA dataset which was removed from the manuscript as suggested by the reviewer.
Comment 7 (lines 114 and 125)
Please note that Table S1, as well as Figures S1 and S2 are not available to the Reviewer. These files from Supplementary are available for viewing only after accepting the article. This complicated the process of reviewing the manuscript.
Answer: We agree with the reviewer. However, we also submitted a file with the supplementary material. Since, as suggested by the reviewer, we removed everything related to a special case study of Portugal, we decided to include in the manuscript the supplementary material, which we believe better illustrates our findings.
Comment 8 (Figure1)
The resolution of Figure 1 is low and does not meet the required resolution of 600 DPI to provide cartographic information in the MDPI Fire Journal. Recall that not all MS Word to PDF converters work well. Please use a professional PDF converter.
To improve the resolution, it is recommended to limit the drawing area to Southern Europe, i.e. to the territory of the mentioned countries Portugal, Spain, France, Italy, and Greece.
Answer: We produced all figures with 600 dpi to meet the required resolution of the MDPI Fire Journal. We can submit files with all the figures with 600 dpi.
Comment 9 (Global and Figure1 continue)
Are Africa, Turkey, Syria and Israel now also part of Europe? Please clarify how the wildfire burned areas and land cover changes were calculated for EU area.
Please make appropriate edits to the text of the manuscript and to Figure 1.
Answer: We agree with the reviewer. We have updated the figure, namely with regard to the study period and explained in the manuscript that the calculations were only carried out for Europe and did not consider fires outside European territory. However, we decided to keep the polygons of the area burned by fires outside Europe, to better illustrate the spatiotemporal distribution of this phenomenon, in particular, the magnitude of the incidence of fire in the countries of Southern Europe
Comment 10 (missing Figure)
In addition to Figure 1, the Reviewer recommends providing the EFFISBA information on changes in the total area over the years, both for individual countries and throughout Europe.
Without this picture, it is impossible to draw any conclusions about the increase in wildfire activity.
Answer: We decided not to show the evolution of the annual burned area in each country and entire Europe in the previous version of the manuscript, mainly because the metrics/indicators we defined and used to access LULCC, including within burnt areas area, are independent of the wildfire activity. However, as suggested by the reviewer, we included a new Figure 3 to illustrate the interannual variability of the burnt area in Europe and in each of the southern countries
Comment 11 (Global and Figure 2)
The general recommendation when showing drawings is to avoid labels such as “left panel”, “central panel”, “right panel”. It is recommended to use letter designations in the Figures, such as: (a), (b) etc. Accordingly, in the text of the manuscript, references to these Figures are given in the form of Figure 2a, Figure 2b etc. Please make appropriate changes to the Figures. Comment 12 (Figure 2, continue, right panel) The expediency of casting in Figure 2 the data of ICNFBA, which is using only in the study case of Portugal, are discussed above, see Comment 2.
Answer: We changed the figures and updated the text accordingly.
Comment 13 (missing Figure)
The manuscript does not contain any maps of land cover. The reader may not know where forests, shrublands and croplands grow in Europe.
Answer: The authors believe that the maps of LULCC in Europe (Fig.1) allow the reader to have an overview of the main LULC, in addition to the changes for the different frame periods. We decided not to include the LULC map essentially for three main reasons: (i) this study is about the relationship between fires and LULCC, and not between fires and LULC; (ii) the LULCC map provides the information about the LULC transitions, that is, it provides the LULC information for a CLC inventory together with the classes of the following inventory; finally (iii) LULC/CLC maps are easily obtainable in numerous publications (e.g., Pereira et al., 2014, cited in the manuscript), on the EEA website (e.g., https://www.eea.europa.eu/data-and-maps/figures /corine-land-cover-types-2006) and on the web.
Comment 14 (abbreviation, line 197)
The longest abbreviations such as CLCCEFFISBA, OEFFISBA, OCLCBA, or EFFISBA are difficult to understand. It is recommended to make appropriate changes.
Answer: We agree with the reviewer and understand his/her suggestion. However, the number of these acronyms was significantly reduced when we removed everything associated with the secondary aim/case study of Portugal. In the current version of the manuscript, beyond well-known acronyms (e.g., CLC, LULC, EFFIS, BA) we only use 3 acronyms (RTLA, RTGA, NCA) to identify the change indicators and one (CLCCEFFISBA) to characterize the LULCC within burnt areas. We hope the reviewer considers this an acceptable number of acronyms. The alternative is to repeat several times the same text (the definition of these indicators).
Comment 15 (Figure 3)
The layout of this Figure is unsuccessful. Figure 3 shows a comparison between the countries of Europe (EU), France (FR), Greece (GR), Italy (IT), Portugal (PT) and Spain (SP), while for the purposes of the manuscript, the study of dynamics in LULC for the period 2000 – 2020 is indicated. The columns in the Figure should be rearranged so that they reflect the dynamics of the processes.
Answer: The amount of information presented in Figure 3 (and Figure 4) is huge. There is no simple way to represent this information without significantly increasing the number of figures. The type of figure we chose intends to provide all the information (results obtained) to the reader in as few figures as possible. This type of graph aims to assess the dynamics of the process because it allows to simultaneously check and compare the results obtained: (i) for each one of the countries in southern Europe and for the whole of Europe; (ii) for the CLC classes that underwent major changes (LULCC) and for each of the 3 subperiods of the study period. We have not found any other way of presenting all these results, which allows a better illustration of the dynamics of the processes. However, we are fully available to rearrange the columns of Figure 3, as the reviewer indicates. In this answer, we refer to the figure numbers in the first version of the manuscript, (initially submitted), not the new/revised version of the manuscript.
It is also important to mention the following: Corine Land cover has 44 different classes of LULC; each of these 44 classes can transition/change to each other or that means a huge number of (almost 2000) possible transitions. This high number of possible transitions constituted a challenge for the presentation of the obtained results. We intend to isolate the signal from the noise. Our decision was based mainly on the objective of the work (quantitatively evaluating the relationship between fires and LULCC) and consisted of representing the transitions grouped at level 2 of the CORINE Land Cover nomenclature. This decision allowed focusing on the LULC classes possibly related to the fires, namely 21 Arable lands, 22 Permanent crops, 31 Forests, 32 Scrub and/or herbaceous vegetation associations, and 33 Open spaces with little or no vegetation. This decision also considered that the transitions between these classes represent the vast majority, namely more than 80% of RTLA and around 80% (on average) of RTGA. This means that: (i) transitions allegedly not potentially associated with fires occur in a much smaller area (in some cases they are almost negligible); (ii) this type of graph should make it possible to show the relationship between fires and LULCC.
Comment 16 (Figure 3 continue)
It is not clear why the Relative Total Lost Area (RTLA, left panel) and Relative Total Gained Arena (RTGA, right panel) should be presented in the same Figure. It is recommended to present these values in two different Figures and make them more readable for perceiving the land cover change dynamics.
Answer: Obviously, we can easily present the two panels of Figure 3 in two different figures. This change will just allow us to present both figures with a larger size. However, we do not believe that the current size of the figures is a factor that prevents the reader from reading and understanding the information provided in Figure 3 (and Figure 4).
Comment 17 (Figure 3 continue, and line 241)
The agricultural areas, which usually designated as croplands, are presented in this work in two categories: Permanent crops and Arable land. Please specify why it is necessity to consider these types of vegetation separately. Further in the manuscript, it is necessary to specify the periods and durations of the vegetative cycle for two types of agricultural land cover, as well as the approximate time of straw recycling and/or burning.
Answer: We used LULC and LULCC from CORINE Land Cover (CLC) database. CLC has a nomenclature with different classes in three levels: 5 classes in level 1, 15 classes in level 2, and 44 classes distributed in three levels. The level 2 class of Agricultural areas include 4 level 2 classes: 21 Arable land, 22 Permanent Crops, 23 pastures, and 24 Heterogeneous agricultural areas. We did not define any class of LULC nor have we grouped any class of LULC/CLC based on any additional criteria. We just limited ourselves to calculating the indicators/metrics for all classes of LULC/LULCC and presenting the results obtained only for the classes that obtained the highest values of the indicators/metrics, as explained in the manuscript.
The CLC class 21 Arable land includes 3 level 3 classes: 211 Non-irrigated arable land, 212 Permanently irrigated land, and 213 Rice fields. The CLC class 22 Permanent crops include 3 level 3 classes: 221 Vineyards, 222 Fruit trees and berry plantations, and 223 Olive groves. We are not experts on this topic, but believe that the periods and durations of the vegetative cycle of these agricultural land covers, even for the same class are very different. Additionally, this study did not aim to investigate the role of periods and durations of the vegetative cycle of LULC types on the relationship between wildfires and LULCC.
Finally, firstly, the approximate time of recycling and/or burning the straw should be a feature of only some CLC level 3 classes, while Figure 3 is about CLC level - 2 classes. Secondly, straw recycling should not correspond to a LULCC. Thirdly, the burning of straw must correspond to the different uses of fire (for cleaning agricultural soil, forestry, renovating pastures, cleaning paths, accesses, and installations, protection against wildfires, etc.) but not as a wildfire.
Comment 18 (Figure 3 and line 241) The CLC33 category (open space with little or no vegetation), which is presented in Figure 3, includes the sparsely vegetated areas (CLC333) and Burnt Area Burnt Area (CLC334). Further the climate and wildfire impacts are described by following CORINE CLC subcategories correspondently: Sparsely vegetated areas (CLC333) and Burnt Area (CLC334). However, the Reviewer did not found any information about the CLC33 category in the manuscript, as well as any discussion of the causes of desertification in Southern Europe (CLC333).
Answer: It is important to note that Figure 3 only illustrates the LULCC in Europe and the countries of the South and not the causes of these changes. Additionally, and as mentioned by the reviewer (comment 5), and recognized by the authors, this study was not about climate change, fire emissions, or causes, nor about effects of climate and desertification, but only about the relationship between fires and LULCC.
It should be noted that CLC33 and its respective subcategories were mentioned several times in the manuscript. Please see, for example, lines 162, 170, 285, 296, 298, 302. In addition, CLC 33 is one of the five level-2 classes that most changed in the study period, as illustrated in Figure 3. However, changes in CLC 33 are very small except in a small number of cases (Figure 3). RTLA was only significant in Greece during the last frame period and RTGA was somehow higher in Greece during the second frame period and in Italy during the third frame period. Changes in CLC33 were despicable when considering the entire European territory and France, in particular. Changes in CLC33 are only relatively higher in Portugal and Spain, which are, by far, the two most affected countries by wildfires, in terms of the total number of wildfires (>53%) and burnt area (>61%). Obviously, these results, especially class 334 Burnt areas, justify the position of CLC33 as one of the 5 classes with the highest LULCC. However, although some regions have high fire recurrence/frequency, in the vast majority of locations the fire return period is longer than the interval between CLC inventories. Additionally, all other subclasses of CLC33 are characterized by non-existence or sparse vegetation, which poses difficulties for the propagation and final size of the wildfires.
However, we agree with the reviewer that these aspects (CLC33, climate, wildfire impacts, desertification) are important and must be discussed in the manuscript. Therefore, we included some text about these topics.
Comment 19 (missing Figure)
The Reviewer did not find a picture showing the results of the analysis of RTLA and RTGA for the second datasets of the European Forest Fire Information System (EFFIS).
Answer: On one hand, the Relative Total Lost Area (RTLA), is defined as the CLCCLA evaluated for each period and each CLC class, divided by the total CLCC area (TCLCC) for the same period. On the other hand, Relative Total Gained Area (RTGA), is defined as the CLCCGA evaluated for each period and each CLC class, divided by the total CLCC area (TCLCC) for the same period. This means that RTLA and RTGA are indicators to assess LULCC, and can only be used to assess changes in LULC datasets.
Comment 20 (Figure 4)
Again, the layout of Figure is unsuccessful. The picture is not readable. In this work the dynamics is investigated, so the columns, for example for France, should be grouped together.
Answer: We agree with the reviewer that the results can be presented differently. As the figure stands, it allows a better assessment of the spatial variability (between countries) in each subperiod. If the figure is produced as suggested by the reviewer, it should allow a better assessment of temporal variability (between periods) in each country. None of the figures will allow an easy comparison of both the spatial and temporal variability. Please our answer comment about Figure 3. We are fully available to change figures 3 and 4 as suggested by the reviewer, but we feel that both ways have the same advantages and disadvantages.
Comment 21 (Figure 4, continue)
The labels such as 311–324 should be avoided. If the result is difficult to present in the figure, then it should be presented in the form of a table.
Answer: We agree that it is not easy to read this figure as it would be not better to read Tables, mostly because of the great diversity of different transitions. We believe that this comment is not about presenting the results in a Figure or a Table but about the labels used to identify the transitions.
These results can be presented in a Figure or in 3 Tables. In fact, Figure 4 is also one way of presenting three figures (one for each period) in just one figure. We believe that results are better presented in just one figure also because color patterns make easier the identification of patterns than values in three different tables.
The difficulty of labeling the transitions does not disappear by replacing the figure with three tables. We can change the way we identify the transitions. For example, we can change “311–324” from “Broad-leaved forest to Transitional woodland-shrub” but we do not believe that is a simplification. We thought about it, but we weren't able to find the simplest/easiest way to identify the transitions. However, we are fully available to change the figure or tables, with a way of identifying the transitions to be indicated by the reviewer.
Comment 22 (Sections 3.3 and 4.3)
The case study of Portugal is recommended to separate into a separate article.
Answer: We remove that part, as suggested by the reviewer.
Comment 23 (Conclusion, line 444–454) It is recommended to remove common phrases, focusing the readers' attention on the main results of the work.
Answer: As many readers only read or start by reading the abstract and conclusions, we decided to start the conclusions with a brief (2 short paragraphs) summary of the manuscript. However, we decide to remove these two paragraphs, as suggested by the reviewer.
Comment 24 (Conclusion, line 452–454)
“To this aim, we defined different indicators allowing us to (i) assess the main land use/land cover changes that occurred in Europe between 2000 and 2020, and (ii) quantify and better understand the relationship between these changes and wildfires.”
The stated time range is 2000–2020, but in the text of the manuscript and in Figures 2, 3, and 4, the range is smaller. Please clarify this issue.
Answer: We agree and thanks the reviewer for the possibility to clarify this issue. We defined the study period as 2000 – 2020 to round the starting and ending years.
The CORINE Land Cover (CLC) inventories were made available in 2000, 2006, 2012 and 2018 but have a minimum time consistency of ± 1 year. This means that the CLC data reflects the LULC state from 1999 to 2019. However, the EFFIS Burnt area dataset starts in 2000 and, currently, accounts BA polygons of 2023 wildfires. Therefore, the starting year of EFFISBA and the CLC first map of 2000 defines the starting year of the study period and the ending of the LULC data defines the ending of the study period in 2019. Following the reviewer comment, we decide to correct the study period to 2000 – 2019. Consequently, we also update the figure with the burnt area polygons in Europe (Figure 2, in the revised version of the manuscript).
It is important to mention that we are particularly interested in the LULCC, are computed between two consecutive CLC inventories, i.e., 2006-2000, 2012-2006 and 2018-2012, well covered by the EFFISBA dataset used in the study.
Comment 25 (Conclusion, line 473–480)
“Besides, a secondary aim was to evaluate the discrepancies between different BA inventories at the national level to justify the adequacy of our methodological decisions. Results revealed a good match between the official European wildfires dataset (EFFISBA) and the national Portuguese BA dataset, suggesting that the first can be used for investigation at the national level in the absence of more reliable local information.”
The second goal requires a separate publication
Answer: As mentioned before, we agreed to remove from the manuscript all parts of the second objective of the initial work, in which a special case study from Portugal is investigated.
Reviewer 4 Report
The submitted manuscript titled "Soil microbial biomass and activity indicators over a range of Mediterranean sites under desertification risk" by Parente et al., assesses a robust set of soil indicators which to assess the transitions between land use/land cover change and burnt area (BA) in Europe in the last two decades (2000 – 2020). This manuscript contains a robust set of variables, which is interesting, but in my opinion, the present version is too long (especially in the result section). The use of too many acronyms makes it sometimes difficult to follow the reading. So, I suggest that major revisions are needed to clarify the manuscript and, in particular, ensure that the conclusions are well-supported by the results.
Specific comments
Line 36-37: This sentence is not clear. Hazardous events i.e wildfire, can affect wildfire? Please clarify.
Line 40-43: Please break down this long sentence into smaller ideas.
Line 43-45: Please clarify “human LULC”; these acronyms appear here for the first time.
Line 63-65: At lines 59-61, the authors underline the lack of knowledge about the relationship between LULCC and wildfire, then at lines 63-64 they suggest that this relationship is “generally recognized”. What is true? Please clarify.
Line 67: Please clarified the term “BA”; this acronym appears for the first time here and was not connected with the general description of wildfire events.
Lines 66-86: Rephrase the aims to betters specify them, avoiding referring to the table. In the present form, the aims are not clear. the scientific question starting at line 73 is your hypothesis? In this case, the authors should rephrase the hypothesis. The hypothesis should have proposed expected results. For example, the first hypothesis/ question reads “What the type of LULC can prevail in Europe in the last two decades? Now the question/ hypothesis is not giving a clear direction of the expected results. This applies to all three hypotheses given in the manuscript.
Material and Methods:
Lines 102- 112: this part is not focused on the description of the methodologic approach used, as a result, this paragraph is very confusing. Probably this part should be synthesized or shifted in the discussion section.
Lines 132-136: This sentence is very long and confusing. What’s the EFFIS map? How this information is helpful in your methodology?
Line 154-187: subsection 2.3 refer only to Portugal. Why in previous sentences (lines 148- 149) the author write “five countries were analyzed”? please clarify this choice
Result and discussion section
For the two sections, I consider that having more specific hypotheses or expectations about the results (in the introduction) would help focus this section on those trends that are supporting or reject the hypotheses. Also, several instances in this section exist where results are described again but not framed or discussed within the proposed objectives or hypotheses. There are some good discussion points raised here that if better justified could enhance the reach of this study. Moreover, these sections are two long, so it is difficult for the reader to take off the message of your study.
Conclusions should be thoroughly supported by the results presented in the manuscripts or
The manuscript requires some language editing to make it clearer and more understandable
The manuscript requires some language editing to make it clearer and more fluent
Author Response
Answers to Reviewer 4
The submitted manuscript titled "Soil microbial biomass and activity indicators over a range of Mediterranean sites under desertification risk" by Parente et al., assesses a robust set of soil indicators which to assess the transitions between land use/land cover change and burnt area (BA) in Europe in the last two decades (2000 – 2020). This manuscript contains a robust set of variables, which is interesting, but in my opinion, the present version is too long (especially in the result section). The use of too many acronyms makes it sometimes difficult to follow the reading. So, I suggest that major revisions are needed to clarify the manuscript and, in particular, ensure that the conclusions are well-supported by the results.
Answer: We are very grateful to the reviewer for the time and effort taken to revise our manuscript as well as for the suggestions for changes that we are sure will improve the quality of our manuscript.
Another reviewer suggested removing from the manuscript all parts of the second objective of the initial work, in which a special case study from Portugal was investigated. We agreed with this suggestion and hope that this change contributes to significantly reducing the size of the manuscript, (especially in the result section), as suggested by the reviewer.
Another reviewer also suggests a reduction in the number of acronyms. We agree with the reviewers and understand their suggestions. However, the number of these acronyms was significantly reduced when we removed everything associated with the secondary aim/case study of Portugal. In the current version of the manuscript, beyond well-known acronyms (e.g., CLC, LULC, EFFIS, BA) we only use 3 acronyms to identify the change indicators (RTLA, RTGA, NCA) and one (CLCCEFFISBA) to characterize the LULCC within burnt areas. We believe that this reduced number of acronyms is more acceptable, especially because the alternative to the use of these acronyms is to repeat several times the same text (the definition of these indicators).
Specific comments
Line 36-37: This sentence is not clear. Hazardous events i.e wildfire, can affect wildfire? Please clarify.
Answer: We used the adjective “hazardous” because wildfires are frequently considered natural and/or human hazards. However, we agree with the reviewer that the word can be confusing. We change the text to clarify the idea.
Line 40-43: Please break down this long sentence into smaller ideas.
Answer: We change the text accordingly.
Line 43-45: Please clarify “human LULC”; these acronyms appear here for the first time.
Answer: We change the text to define the acronym LULC.
Line 63-65: At lines 59-61, the authors underline the lack of knowledge about the relationship between LULCC and wildfire, then at lines 63-64 they suggest that this relationship is “generally recognized”. What is true? Please clarify.
Answer: We agree with the reviewer that this apparent contradiction needs to be clarified. Whether the dependency relationship between fires and LULCC is generally recognized by researchers, has yet to be quantitatively established. We have changed the text to clarify this apparent contradiction.
Line 67: Please clarified the term “BA”; this acronym appears for the first time here and was not connected with the general description of wildfire events.
Answer: The acronym BA is defined in line 48.
Lines 66-86: Rephrase the aims to betters specify them, avoiding referring to the table. In the present form, the aims are not clear. the scientific question starting at line 73 is your hypothesis? In this case, the authors should rephrase the hypothesis. The hypothesis should have proposed expected results. For example, the first hypothesis/ question reads “What the type of LULC can prevail in Europe in the last two decades? Now the question/ hypothesis is not giving a clear direction of the expected results. This applies to all three hypotheses given in the manuscript.
Answer: As explained before (answer to first comment) another reviewer (reviewer#2) suggested removing from the manuscript all parts of the second objective of the initial work, in which a special case study from Portugal is investigated. Therefore, the text in lines 68-72 was removed, which included the reference to Table 1.
We change the manuscript to clarify the research questions, hypothesis, objectives, goals and expected results.
Material and Methods:
Lines 102- 112: this part is not focused on the description of the methodologic approach used, as a result, this paragraph is very confusing. Probably this part should be synthesized or shifted in the discussion section.
Answer: We change the manuscript accordingly. This text is now subsection 4.1 of the discussion.
Lines 132-136: This sentence is very long and confusing. What’s the EFFIS map? How this information is helpful in your methodology?
Answer: The instructions for Authors of Fire MDPI recommend the author to described with sufficient detail the materials and methods to allow others to replicate and build on published results. We consider it important to inform the reader about the database development process and spatial resolution because can influence the obtained results. However, following the reviewer's suggestion, we reduced the description and amount of text.
Line 154-187: subsection 2.3 refer only to Portugal. Why in previous sentences (lines 148- 149) the author write “five countries were analyzed”? please clarify this choice
Answer: As explained before (answer to first comment) another reviewer (reviewer#2) suggested removing from the manuscript all parts of the second objective of the initial work, in which a special case study from Portugal is investigated. Therefore, this text was removed.
Result and discussion section
For the two sections, I consider that having more specific hypotheses or expectations about the results (in the introduction) would help focus this section on those trends that are supporting or reject the hypotheses. Also, several instances in this section exist where results are described again but not framed or discussed within the proposed objectives or hypotheses. There are some good discussion points raised here that if better justified could enhance the reach of this study. Moreover, these sections are two long, so it is difficult for the reader to take off the message of your study.
Answer: As mentioned before, in answer to another comment from the reviewer, we change the manuscript to clarify the research questions, hypothesis, objectives, goals, and expected results.
Conclusions should be thoroughly supported by the results presented in the manuscripts or
Answer: Conclusions should be thoroughly supported by the results presented in the manuscript.
The manuscript requires some language editing to make it clearer and more understandable
Answer: The revised version of the manuscript underwent a new language edition to make it clearer and more understandable.
Reviewer 5 Report
The paper presents the contents in an efficient and organized manner. The paper begins with a brief introduction that gives a general overview of the topic.
It is recommended to review the formatting of the text, the hyphenation and the word spacing.
It is suggested to increase the bibliography regarding the LULC classification methodology, with which the data used for the statistics reported within the paper have been compiled. With regard to the case study of Portugal, it would be useful to better discriminate the areas that have been subjected to the highest loss of vegetation cover due to wildfire, including through maps to specify geographically where the major loss is concentrated in recent years.
the English language level of the paper is appropriate and clear
Author Response
Answers to Reviewer 5
The paper presents the contents in an efficient and organized manner. The paper begins with a brief introduction that gives a general overview of the topic.
Answer: We are very grateful to the reviewer for the time and effort taken to revise our manuscript as well as for the suggestions for changes that we are sure will improve the quality of our manuscript.
It is recommended to review the formatting of the text, the hyphenation and the word spacing.
Answer: The revised version of the manuscript underwent a new language edition to make it clearer and more understandable, and solve some typos.
It is suggested to increase the bibliography regarding the LULC classification methodology, with which the data used for the statistics reported within the paper have been compiled. With regard to the case study of Portugal, it would be useful to better discriminate the areas that have been subjected to the highest loss of vegetation cover due to wildfire, including through maps to specify geographically where the major loss is concentrated in recent years.
Answer: We increase the bibliography regarding the LULC classification methodology, with which the data used for the statistics reported within the paper have been compiled.
Another reviewer suggested removing from the manuscript all parts of the second objective of the initial work, in which a special case study from Portugal is investigated. Therefore, all the text associated with the case study of Portugal was removed.
Round 2
Reviewer 4 Report
The manuscript "Quantitative assessment of the relationship between land 2 use/land cover changes and wildfires in Southern Europe" is now improved.
This form is suitable for publication